# Green urbanization

Jan Eeckhout[1]*, Christoph Hedtrich[2]

**1** Department of Economics and ICREA-GSE-CREi, Universitat Pompeu Fabra, Barcelona, Spain,
**2** Department of Economics, Uppsala University, Uppsala, Sweden

* jan.eeckhout@upf.edu

## Abstract

Large cities are more productive and generate more output per person. Using data from the UK on energy demand and waste generation, we show that they are also more energy-efficient. Large cities are therefore greener than small towns. The amount of energy demanded and waste generated per person is decreasing in total output produced, that is, energy demand and waste generation scale sublinearly with output. Our research provides the first direct evidence of green urbanization by calculating the rate at which per capita electricity use and waste decrease with city population. The energy demand elasticity with respect to city output is 83%: as the total output of a city increases by one percent, energy demand increases less than one percent, and the Urban Energy Premium is therefore 17%. The energy premium by source of energy demand is from households (13%), transport (20%), and industry (16%). Similarly, we find that the elasticity of waste generation with respect to city output is 90%. For one percent increase in total city output, there is a less than one percent increase in waste, with an Urban Waste Premium of 10%. Because large cities are energy-efficient ways of generating output, energy efficiency can be improved by encouraging urbanization and thus green living. We perform a counterfactual analysis in a spatial equilibrium model that makes income taxes contingent on city population, which attracts more people to big cities. We find that this pro-urbanization counterfactual not only increases economic output but also lowers energy consumption and waste production in the aggregate.

**Data Availability Statement:** We have uploaded all data and instructions to replicate our findings to zenodo.org. This includes the raw data, the final dataset used for analysis and the programs written to perform the analysis. The assigned DOI is: 10.5281/zenodo.5167033.

## Introduction

Urbanization is an irreversible and ongoing force in industrialized and developing countries alike. Around the world, more than half the population lives in cities and urban areas, and in several countries like the UK or the Netherlands, this fraction is over 70% and growing [1]. The ecological implications are far-reaching. Large cities use a lot of energy, they have a lot of traffic and pollution, and they produce a lot of waste. Beijing, London, New York, and Bombay have long been known as much for their dirty alleys as for their scarce green spaces. Wandering around Bethnal Green in London, Charles Dickens' character Fagin', Oliver Twist's petty criminal trainer "soon became involved in a maze of the mean dirty streets which abound in that close and densely-populated quarter." [2].

**Funding:** This study was funded by financial support from the European Research Council Advanced Grant (339186) and MICINN (AEI/FEDER, UE-PGC2018-096370-B-I00) awarded to J. E. The funders had no role in study design, data collection, and analysis, decision to publish, or preparation of the manuscript.

**Competing interests:** The authors have declared that no competing interests exist.

Instead, living in the countryside and in less urbanized areas is considered the ultimate green living experience. With continuing urbanization, this may be a cause for concern and may require immediate action. Yet, from an environmental viewpoint, what matters is not how green a given city is, but how much pollution and dirt all cities *jointly* produce. Big cities concentrate a large number of people in a small space, and twice the number of people in the same space will be more polluting than just once. Therefore, we ask whether energy usage *per capita* is higher in big cities. In other words, to minimize overall energy usage of a system of cities, the question is whether to concentrate people into few big cities or spread population across many intermediate-sized cities.

Recently, a parallel has been identified between cities and biology [3]. In biology, Kleiber's law relates the energy consumption of animals and plants to their mass, and finds a systematic relation of 3/4: as an animal's mass (say in kilograms) increases by one unit, its energy intake increases by 75%. This can be explained by the fractal nature of these organisms, see [4]. Likewise, for cities, different characteristics are shown to be power-law functions of the city population with scaling exponents $\beta$ that fall into distinct universality classes: increasing returns (wealth creation, innovation,. . .) or decreasing returns (infrastructure,. . .). See also [5].

We build on these insights from biology and urban studies with tools from economics. Rather than considering the relation of energy or pollution with population, we consider the relation with *economic productivity*. Our contribution is threefold. First, we provide novel direct evidence of the relation between city output and energy use and waste generation. We find that the elasticity of energy consumption with respect to city output is 83%. In other words, larger cities use substantially less electricity per unit produced. And the elasticity of waste generation with respect to city output is 90%. We thus find that large cities are more energy efficient, which is in line with arguments that laud the virtues of dense urban areas for ecological efficiency [6].

Our second contribution is to focus on the energy-efficient production of output: How should the population spread out over all cities to generate the most economic output, with the least use of energy resources? In particular, we analyze the urban population distribution as an equilibrium outcome where citizens' behavior and location choices affect the supply and demand of labor as well as prices. When the Bay area in the US saw an increase in productivity due to the tech boom, wages increased and people moved there to find jobs thus increasing the local population. The urban population equilibrium with mobile workers is a key determinant of energy efficiency because it affects the density of population, which in turn is an important determinant of energy consumption and waste production.

Third, we show how a tax schedule that modifies the existing labor income tax system generates energy efficiency gains. The central premise is that people trade-off higher earnings from labor with the cost of living, which leads to an equilibrium population distribution. The change to the tax system has positive effects for both economic output and pollution, i.e. while total economic output would increase, total energy demand would fall. This is important because many proposals to foster sustainability lead to a trade-off between material wealth and the protection of environmental resources. Instead, we show that we can obtain *both* higher output and better environmental protection. That is the main insight of green urbanization.

## Wage, energy and waste premia of urbanization

### Data and analysis

Cities are identified as travel-to-work areas (TTWAs), as defined by the ONS in 2011. There are 218 TTWAs in Great Britain, covering the full landmass of the country. They are intended to represent labour market zones: their boundaries have been drawn to ensure that 75% of the

residents of each TTWA work in the same one, and that 75% of those who work in each TTWA live in the same one. However, most of the local data used is based on Local Authority Districts and Unitary Authorities (LAs), of which there are 380. In many cases though, the boundaries of TTWAs and LAs do not match exactly. The crosswalk between TTWAs and LAs is constructed based on Lower Layer Super Output Areas (LSOAs). For each LA we calculate the share of LSOAs within the LA that belong to a TTWA and use that share as weights when constructing TTWA level statistics. This procedure allows us to construct TTWA level statistics on energy usage and labor market outcomes for 210 TTWAs. The unidentified TTWAs are small and thus represent a very small share of the population.

Local earnings data is based on all workers from the Annual Survey of Hours and Earnings (ASHE), extracted from Nomis. The ASHE is a 1% sample of employees from the HM Revenue & Customs PAYE records. Nomis provides information, by LA, on mean earnings and various percentiles. Total local employment is provided by the Annual Population Survey. The local wage bill is calculated as the product of the average local wage and total local employment.

Energy data is taken from the UK Department of Energy and Climate Change. All energy measures refer to total final energy consumption, see [7]. We use gigawatt hours (Gwh) for our measure of Energy. Local waste data is only available for England and is taken from [8], the web-based system for municipal waste data reporting by UK local authorities to the government. English unitary authorities take care of both disposal and collection. In England, where there is no unitary authority, LAs only have responsibility for collection, and separate "disposal authorities" are responsible for disposal (each disposal authority takes care of the waste from multiple collection authorities). To avoid double-counting we assign all waste based on the collection and unitary authorities and drop the disposal authorities from the data.

All data in the main part of the paper refer to 2010. The waste data is not available in more recent years, but the remaining analysis has been repeated for subsequent years and the results hold also in later samples.

The main empirical results establish the scaling of several city level outcomes. We estimate the scaling coefficient $\beta$ to describe the relationship between two variables $Y$ and $X$, e.g. city energy usage and city population. Further, we assume the data is generated with i.i.d. noise

$$Y_i = X_i^{\beta} e^{\epsilon_i}. \tag{1}$$

To estimate $\beta$ we take logs of this equation and use OLS. For all regressions we weight each observation, a city, by its population.

## The urban wage premium

Large cities are more productive. The value of goods and services generated per person increases in city population [9, 10]. The higher productivity of cities has been observed since Adam Smith. Empirically, it was more formally established first in [11, 12] who discuss the urban-rural wage differential during the great depression of the 1930s. The urban-rural wage differential is generally interpreted as a productivity gap, as in a market economy wages constitute a measure of productivity. The facts are striking and robust across time and space [9, 10]. Let the total wage bill in a city $i$ be denoted by $W_i$, which is the sum of all wages paid out. The urban wage premium is captured by the relation $W_i = W_0 S_i^{\beta}$, where $W_0$ is a constant, $S_i$ is the population of that city, and $\beta$ is the elasticity that measures how wages change in population. An estimated value of $\beta > 1$ indicates that wages and productivity per capita are higher in larger cities.

Fig 1 shows the relation for each city between total productivity as measured by total wages and the population. The estimated scaling coefficient of productivity on city population is

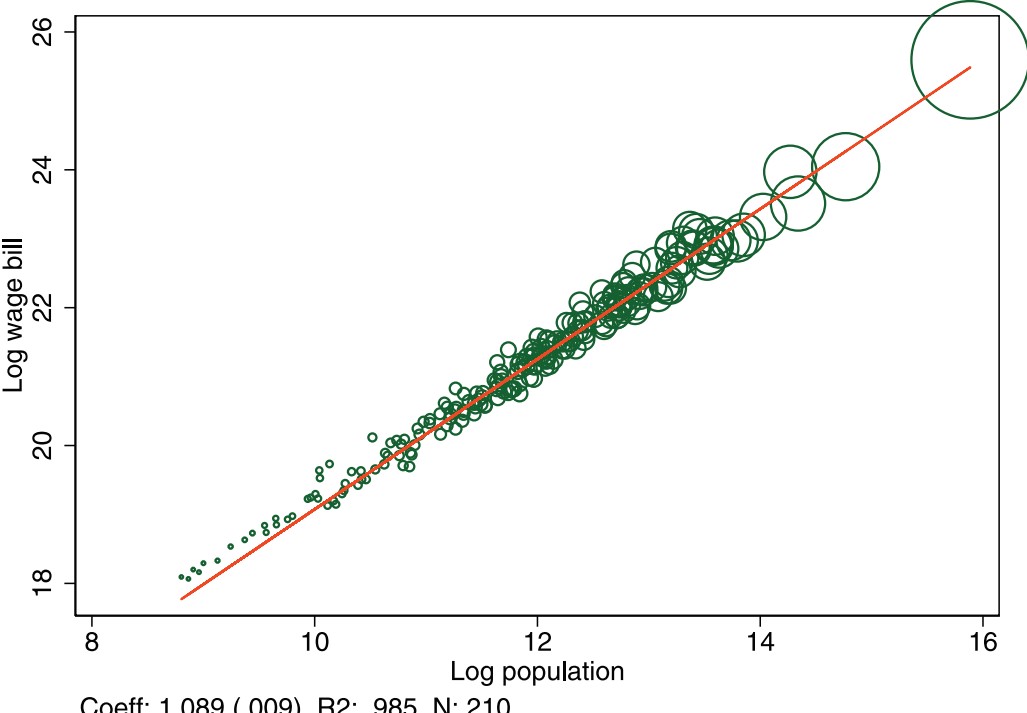

**Fig 1. Urban wage premium.** Total productivity as measured by the wage bill and its scaling with respect to city population. Each dot represents a UK Travel to work area. The scaling coefficient was estimated by OLS and the standard error is shown in parentheses.

$\hat{\beta} = 1.089(.009)$. Comparing a city of 10 million (similar to London, with a population of 8.8 million) with one of 100,000 inhabitants (similar to Dover), the productivity per worker is predicted to be 51% higher in the large city. Several explanations for why workers are more productive in larger cities have been proposed and tested, this includes for example knowledge spillovers between firms, education spillovers between workers, learning, labor market externalities, or network effects, see among others [13–15].

**The urban energy premium.** Big cities not only are more productive, they are are also much more densely populated. And therefore they consume more total energy. To analyze how energy demand changes as more output is produced, we use data obtained from the UK Department of Energy and Climate Change that measures energy consumption at the local level. While there are plenty of aggregated energy datasets, city level information is hard to come by. A notable exception is [16], who use a dataset on a number of cities spread across the globe.

First, our interest is on energy demand per unit of output produced, as measured by wages. This measures energy use in efficiency units, precisely the measure we need to compare across different locations. That is, the optimal location allocation of individuals across space must weigh the cost associated with the energy demand in a city against the benefit from productivity in that city. This expression in efficiency units allows for immediate comparison of the opportunity cost of alternative location decisions.

We find that the elasticity of total energy consumption with respect to total production is 0.83, or an urban energy premium of 17%, as shown in Fig 2. To see how big the savings are, compare a city with a population of 10 million a city with a population of 100,000. The energy

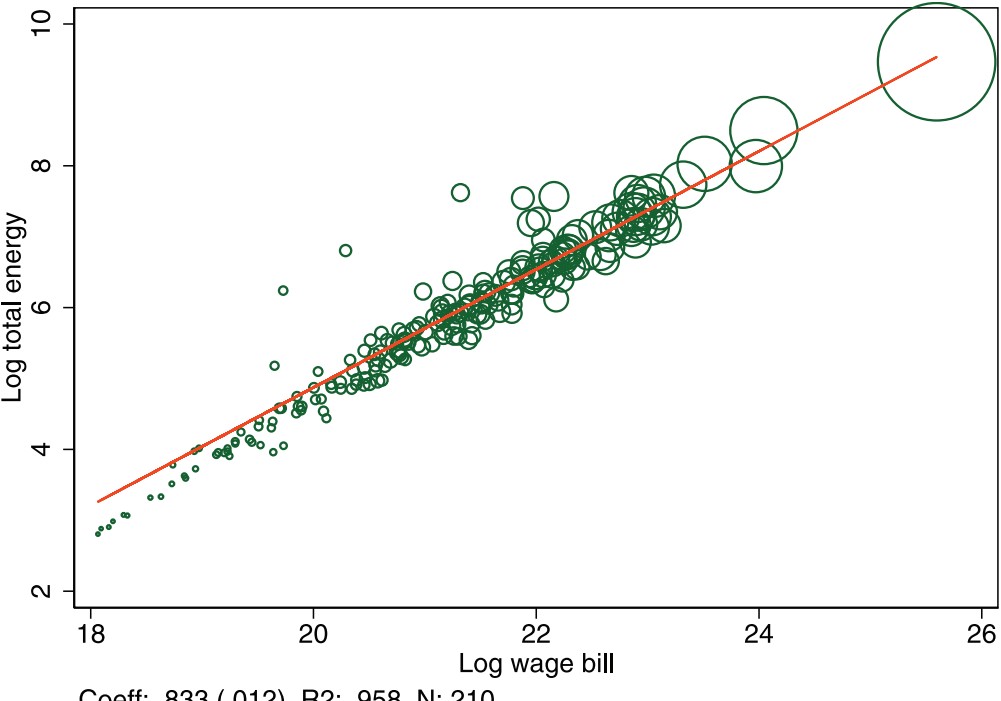

Coeff: .833 (.012), R2: .958, N: 210

**Fig 2. Urban energy premium.** Total energy demanded and its scaling with respect to total wage bill. Each dot represents a UK Travel to work area. The share of total energy demand is 33% for Households, 29% for Transport, and 38% for Industry. The scaling coefficient was estimated by OLS and the standard error is shown in parentheses.

used to produce one unit of output in the big city is a mere 45% of the energy used in the small city.

Instead, the elasticity of energy consumption with respect to population, is 0.92 (.011). Even though output produced per person is increasing in population, energy usage per person is decreasing in population, as the energy efficiency gains more than compensate for additional production.

We can decompose the urban energy premium into the sources of the demand for energy. Of the country-wide energy demand, 38% is industrial, 33% is household demand and 29% is energy demanded for transport. In Fig 3 the elasticity of energy demand with respect to the total wage bill is shown separately by its components: household energy 0.87 (0.008), transport energy 0.8 (0.011), and industrial energy 0.84 (0.021). All coefficients are significantly different from one, and they are stable across a variety of estimation specifications and years. Both household and industrial energy demand drop substantially as city population increases, that is an urban energy premium of 13 and 16% respectively. However, the largest premium at almost 20% is for transport energy demand.

We find strong and significant evidence of lower energy demand in large cities. There are two important determinants of the demand for energy by households: the cost of space and the efficient use of energy per unit of space. The cost of space is further dealt with below where we analyze the role of prices in further detail. Given the higher cost of housing, people in big cities choose to live in smaller spaces. The amount of energy consumption (heating, air conditioning,. . .) decreases with the size of the living space. As a result, those living in big cities consume less energy. Second, energy use of living spaces is more efficient in large cities. Apartment buildings are remarkably energy-efficient structures to live in. Living spaces are insulated

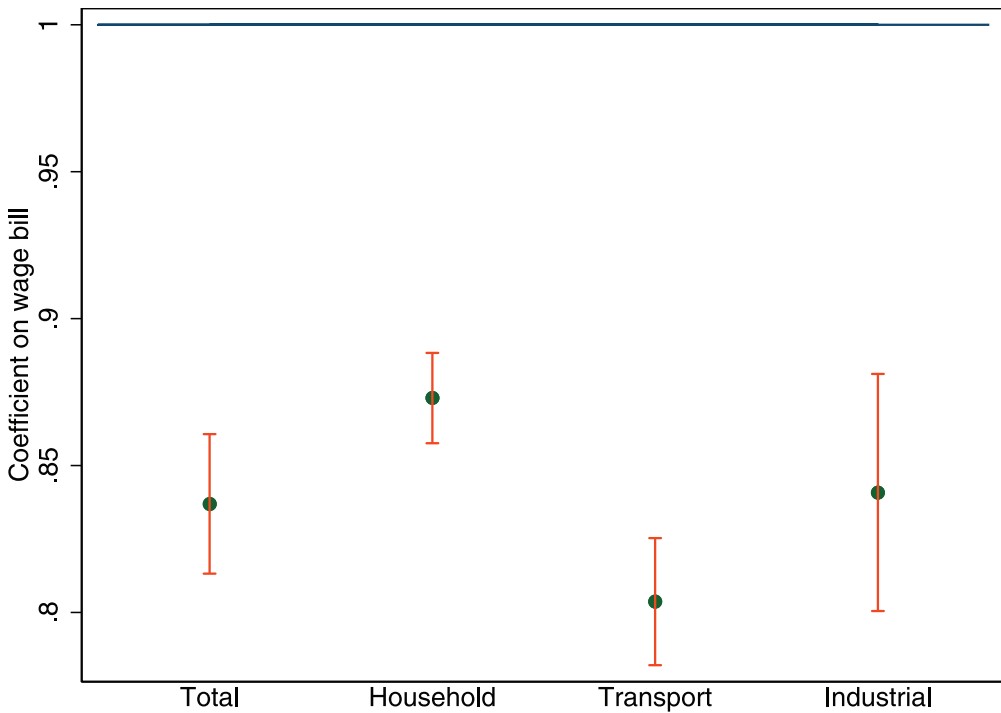

**Fig 3. Urban energy premium by source of energy demand.** The scaling of total energy demanded with respect to the total wage bill by source of energy demand. Each dot represents the estimated scaling coefficient for UK travel to work areas. The scaling coefficients were estimated by OLS and the 95% confidence intervals are shown as red bars.

by adjacent apartments, instead of losing energy through outer walls. Similarly, most apartments do not directly have a roof on top, avoiding large amounts of heat and cold air loss from roof surfaces. Tall apartment buildings have a roof surface per capita that is extremely low, making them very energy efficient.

Lower demand for industrial energy may, at first sight, appear quite normal. Maybe big cities have less heavy industry and that may induce less demand for energy. For example, routine manual jobs are biased towards small cities [17], but it is not clear that those jobs are systematically in industries with higher energy demand. In fact, industry composition does not vary systematically with city population [18]. This suggests that, also for industry energy demand, the cost of space is likely a major driver that leads to more efficient use of energy.

A priori, also the impact of city population on the demand for transportation energy is not immediately clear. On the one hand, commuting distances in large cities are longer, and therefore there is more consumption of energy to get to work. Moreover, there is also more congestion and times may be more than proportional to distance in large cities, further contributing to transport energy demand. But on the other hand, large cities have more (energy) efficient means of transportation, in large part in response to the congestion that population density entails. More people use mass transport—in New York city even the mayor goes to work on the subway –, go by bicycle or go on foot. As a result, the demand for transport energy is lower. Instead, in rural areas without frequent public transportation, families need multiple cars to go to work and to take children to school and extracurricular activities. Only if population density is sufficiently high, mass transit becomes sustainable and walking and bike use picks up. What the data shows is that the second effect strongly dominates since the transportation demand is a major contributor to the Urban Energy Premium.

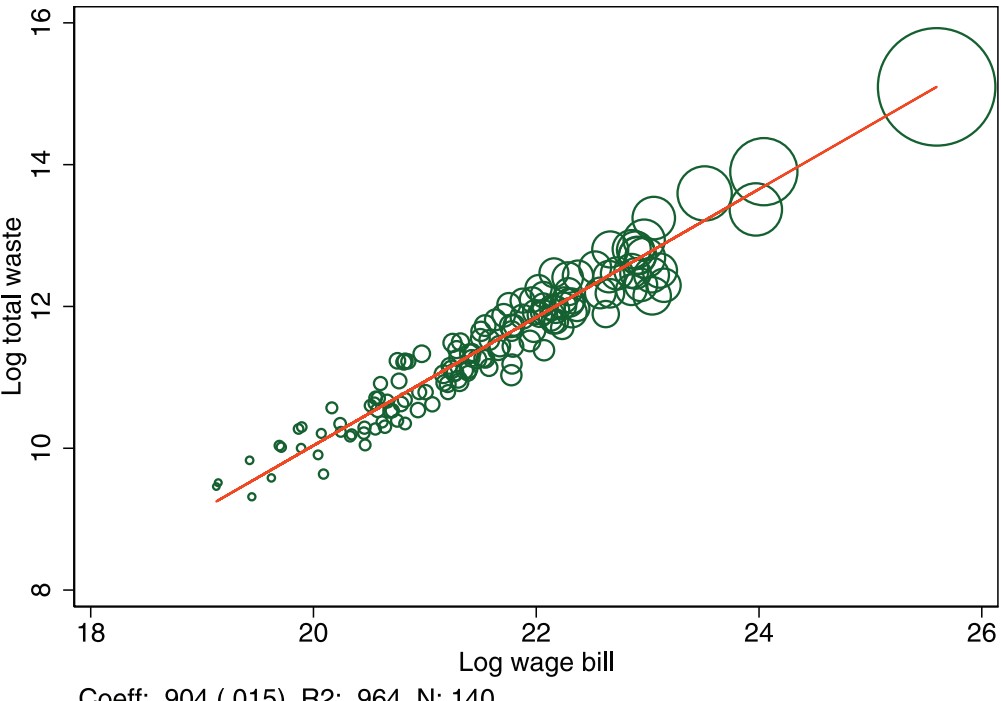

Coeff: .904 (.015), R2: .964, N: 140

**Fig 4. The urban waste premium.** Total waste generated and its scaling with respect to total wage bill. Each dot represents a UK Travel to work area. The scaling coefficient was estimated by OLS and the standard error is shown in parentheses.

**The urban waste premium.** We obtain data from the UK government on waste collection by local authorities. All observations are expressed in Tonnes. Of total waste collected by the UK local authorities, the big majority is household waste (89.6%). Of all waste collected, 35.5% is recycled.

As with energy, we focus on the generation of waste per unit of output. This gives us a measure waste generation in efficiency units and allows us to compare the opportunity cost across different locations. The elasticity of total waste generation with respect to total production is 0.9, or an urban waste premium of 10% (Fig 4). A city with a population of 10 million generates only 64% of the waste per unit of output produced compared to a city with a population of 100,000.

The urban waste premium of 10% relative to the wage bill also implies that the scaling of the total waste produced with respect to population is approximately linear. The major share of the urban waste premium is thus related to the productive efficiency of cities. The overall share of waste generation by source is shown in Table 1. In Fig 5 we further show the urban waste premium by its components: The premium is 13% for household waste and non-household waste, which does not carry a premium, on the contrary, it is negative at -18%, as non-

**Table 1. Waste supply by source.**

|  | Household | Non-household | Total |
| --- | --- | --- | --- |
| Recycled | 33.2% | 2.3% | 35.5% |
| Non-recycled | 56.4% | 8.1% | 64.5% |
| Total | 89.6% | 10.4% | 100% |

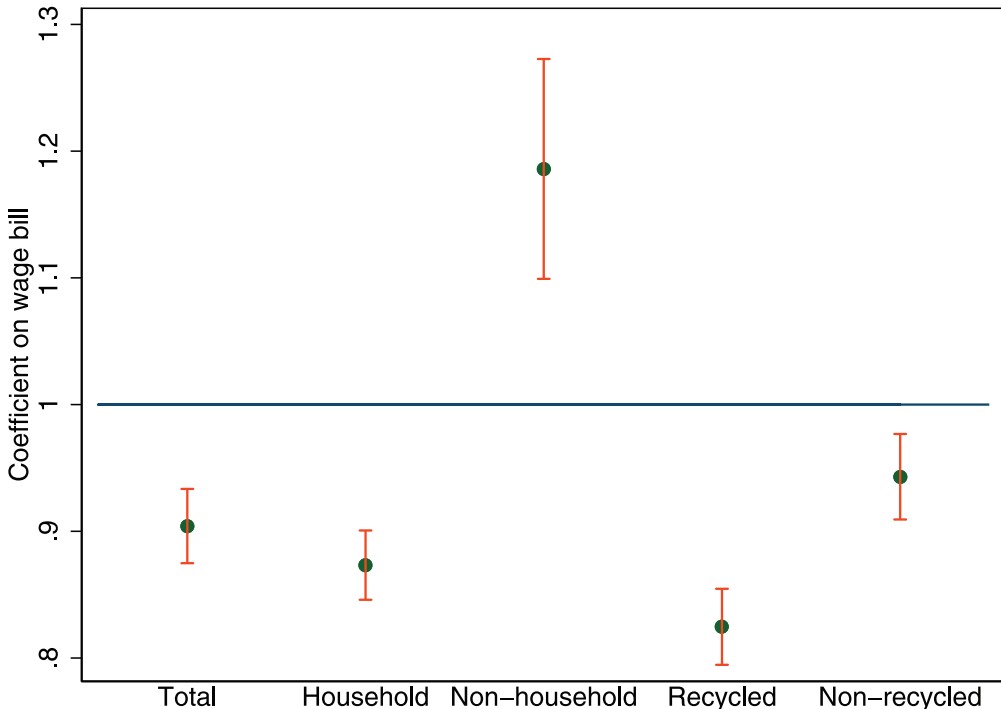

**Fig 5. The urban waste premium by source of waste supply (household vs. non-household) and by destination (recycled vs. non-recycled).** The scaling coefficients were estimated by OLS and the 95% confidence intervals are shown as red bars.

household waste consumption increases more than one-for-one with output. Recycled waste carries the biggest premium of 18% while non-recycled waste has a 6% premium. Households are cleaner in big cities than non-households, and residents of big cities recycle a smaller share of the total waste than those in small cities.

As with energy, housing size is likely a key contributor to waste generation by households. If there is no space in garages and basements, people end up buying less durable goods such as furniture, and home appliances, and they use less home decoration such as carpets and curtains. Parents do not buy climbing racks for the garden in big cities. Periodically all these are thrown out and renewed. However, residents of big cities may spend less time in the house and generate less waste from food and related consumption. This potentially can explain why non-household waste is higher in big cities. Nonetheless, there is a significant urban waste premium as production of waste per unit of output is declining in the population of a city.

Finally, Figs 6 and 7 report the ranking cities in terms of energy and waste efficiency, relative to the population-weighted average (normalized to 100). The larger cities tend to be more efficient, and there is a remarkable variation in the efficiency level. The most energy efficient cities are over 4 times as efficient as the least efficient ones, and the most waste efficient cities are about 3 times as efficient as the least efficient ones.

At this point, it is important to point out that while the environmental impact of one person in a bigger city is smaller, there are markedly negative effects of living in big cities. There are plenty of examples of big cities with poor air quality and other negative features such as pollution, health issues (higher incidence of child asthma), congestion, inequality, or higher temperatures [19]. Cities respond to these negative externalities, for example by emitting less carbon dioxide, either voluntary or imposed by regulation. Los Angeles took strict measures to reduce

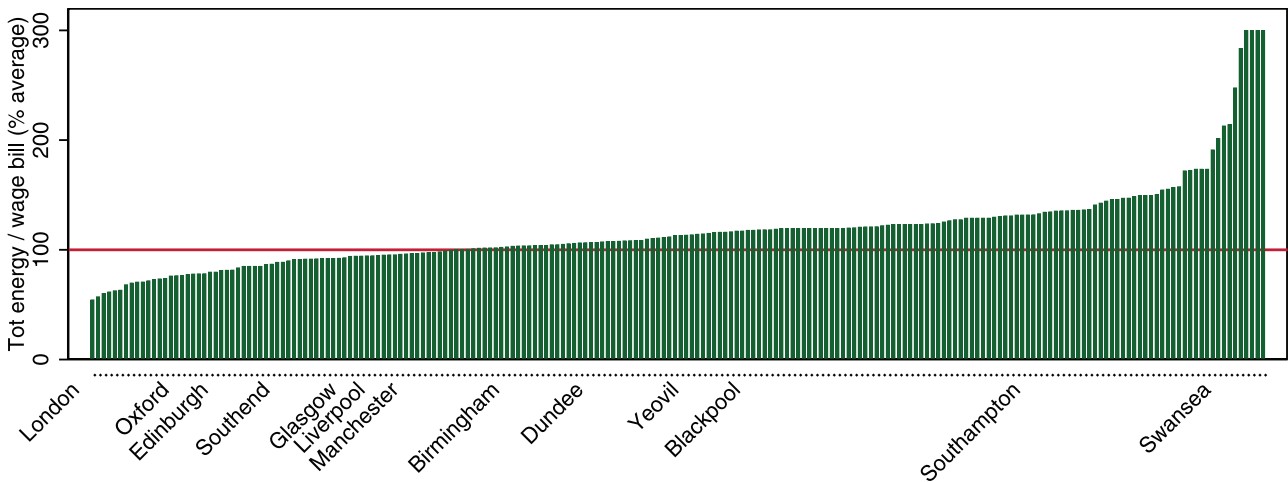

**Fig 6. Ranking of cities by energy efficiency.** Cities are defined as UK travel to work areas.

smog from vehicular emissions, and in London, wood-burning stoves were banned when smokeless zones were legally enforced starting with the Clean Air Act in 1956.

Of course, households may take those negative amenities into account, as they take into account positive amenities in urban areas such as the availability of cultural and entertainment activities, public transportation networks,... In the model below, we allow for those amenities which households take into account when choosing where to locate. The premise of our analysis is that people freely choose where to live trading off the benefits (higher wages, positive amenities) against the costs (housing costs, negative amenities). If so many live in London, it cannot be that the costs outweigh the benefits. Moreover, the evidence in the literature shows that total amenities are independent of city population [20]. That is, there are big variations in amenities, but those variations are not systematically related to the population of a city.

## Counterfactual: Population-specific income tax

We consider a change to income taxation, such that citizens with the same income adjusted for local prices pay the same tax rate in any city. How does this change to the tax system impact

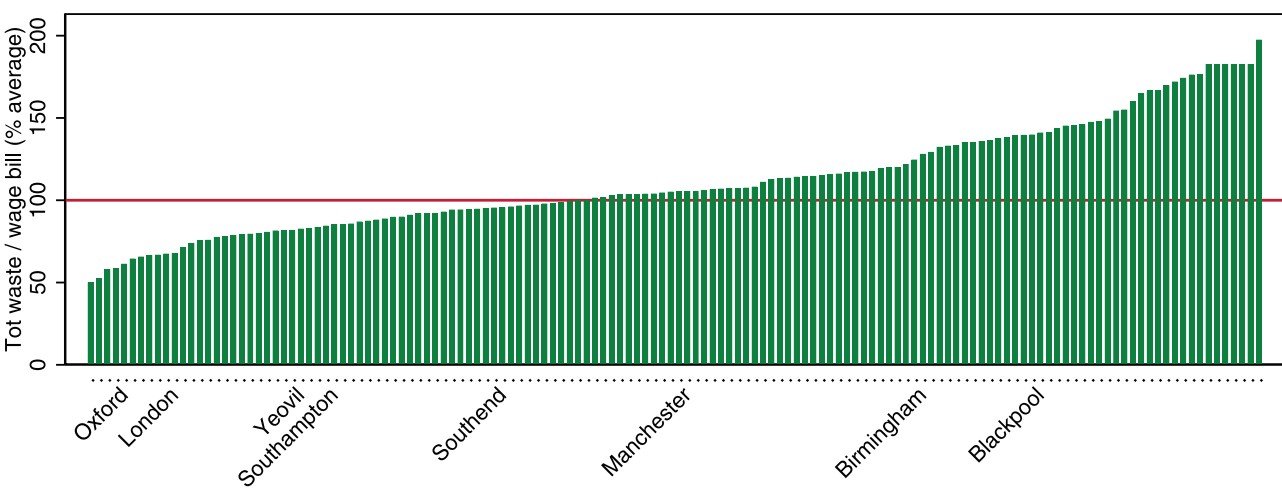

**Fig 7. Ranking of cities by waste efficiency.** Cities are defined as UK travel to work areas.

individuals location decisions, and ultimately also energy and waste consumption? We consider this counterfactual within a spatial equilibrium model.

## The spatial equilibrium model and tax counterfactual

In the entire economy, citizens are mobile and choose the city where to live. Some cities are more productive and pay higher wages, thus attracting more workers. Still, the entire country does not congregate in the most productive city, because space is limited. In particular, housing prices across different locations act as the equilibrating force of population mobility. To be able to perform a taxation counterfactual regarding the impact of income taxation on location choices and ultimately the energy efficiency of the whole economy, we use an equilibrium location choice model. Location choices are a function of wages and house prices, which are determined endogenously.

Progressive labor income taxation that is not city-specific affects workers with the same real income in different cities differently. This is because citizens make their location decision based on utility, which is determined by both post-tax income as well as housing prices. Since income is taxed irrespective of the cost of housing, those living in large cities who earn higher wages and face higher housing costs also pay higher taxes for the same real income, whenever taxes are progressive. We write the post-tax income as $\tilde{w}_i$, and under progressive taxation higher incomes pay a higher percentage of taxes. Empirically, the progressiveness of the tax schedule is well represented by the relation between pre-tax income $w_i$ and post-tax income $\tilde{w}$ as $\tilde{w} = \lambda w^{1-\tau}$ where $\lambda$ is the level of taxation and $\tau$ indicates the progressivity ($\tau > 0$) [21]. The tax rate is proportional and equal to $1 - \lambda$ if $\tau = 0$. Based on past estimates, we assume $\tau = 0.2$.

**Spatial equilibrium model.** Let there be $i = 1, \ldots, N$ cities that differ in their Total Factor Productivity (TFP) denoted by $A_i$. Cities with different productivities coexist and TFP evolves over time satisfying Gibrat's law [22–25]. See Fig 8 for the size distribution of cities (UK Travel to work areas).

At any given time, output produced in a city $i$ with a labor force $l_i$ is $A_i l_i^\gamma$ where $0 < \gamma < 1$ reflects decreasing returns to scale: as the labor force increases, the marginal product of a worker decreases. In a competitive labor market, firms pay wages $w_i$, that are equal to the worker's marginal product: $w_i = A_i \gamma l_i^{\gamma-1}$. Citizens have preferences over the amount of consumption goods $c$ and over the amount of housing $h$, expressed in square meters for example. Denote the utility function by $u(c, h) = \varepsilon_i c^{1-\alpha} h^\alpha$. The noise term $\varepsilon_i$ is a city-specific term that captures measurement error or unobserved city level heterogeneity, for example due to differences in amenities. The noise term will account for the non-systematic variation between the observed outcomes and the model predictions. For the analysis in this paper we take those unobserved differences as given, thus all results we present have to be considered as conditional on the noise term $\varepsilon_i$. However, theoretically is not entirely clear which direction those unobserved factors, if endogenous, would adjust. For example, it could be that negative externalities related to density, like increased travel times and worsening air quality dominate, or that higher density allows for improved amenities for entertainment. However, even if our results quantitatively do not take this into account, as long as citizens take those factors into account when making location decisions, the qualitative result is unchanged. Any citizen chooses in which city $i$ to live, and maximizes utility subject to the budget constraint $c + p_i h = \tilde{w}_i$ where $p_i$ is the price of housing per square meter, the price of consumption is normalized to one, and $\tilde{w}_i$ is after tax earnings. Finally, let the amount of land be fixed and given by $H$. Energy consumption $E_i$ and Waste production $F_i$ in city $i$ depend on the productivity total output $W_i$. Based on our estimates of the Urban Energy and Waste Premium above, we assume

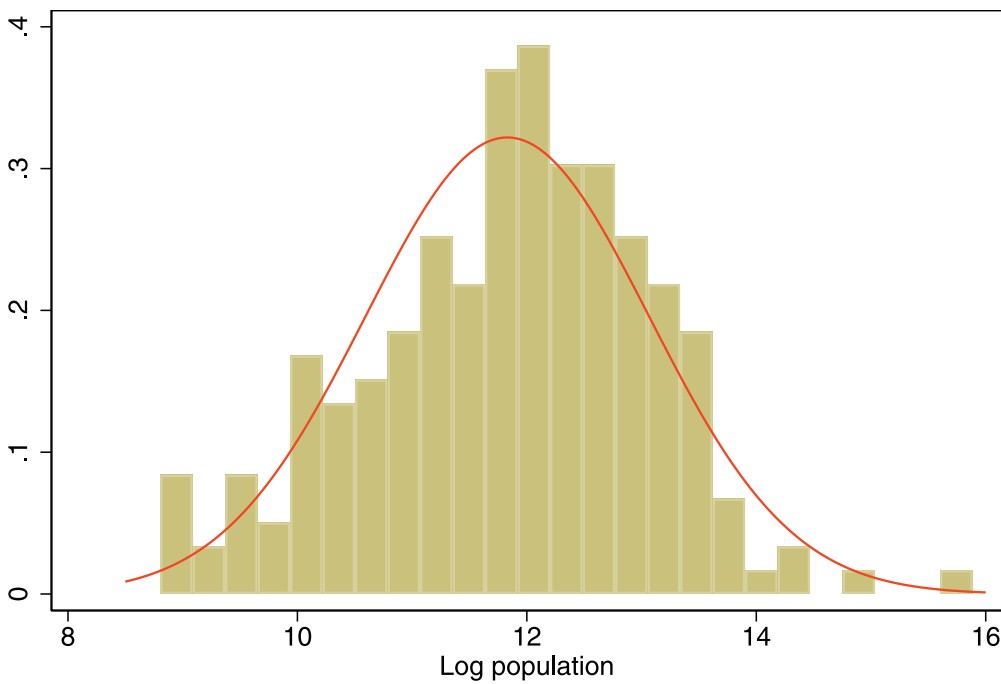

Normal fit: mean 11.83, sd 1.239

**Fig 8. The size distribution of cities (UK travel to work areas).**

that energy demand relates to production $W_i$ as $E_i = E_0 W_i^{0.862}$ and waste production as $F_i = F_0 W_i^{0.905}$ (where $E_0, F_0$ are constants and $W_i = S_i w_i$).

The key feature is labor mobility. Citizens locate where they obtain the highest utility given wages and housing prices. Therefore, the condition that pins down equilibrium is where utility is equalized across cities. The factor $\alpha$ denotes the expenditure share on housing. It has been established that average total expenditure on housing is constant across cites of different population [26]. The housing expenditure of a household in a given city $i$ is $p_i h_i$. The expenditure can be written as a constant fraction $\alpha$ of income: $p_i h_i = \alpha \tilde{w}_i$.

From firm optimization, the first order condition of production is written as $w_i = \gamma A_i l_i^{\gamma-1}$, which is determined by gross or pre-tax income. The first order conditions of consumption imply that optimal housing consumption is $h_i = \alpha \frac{\tilde{w}_i}{p_i}$ and $c_i = (1-\alpha)\tilde{w}_i$, which both depend on post-tax income. Together with market clearing in the housing market, $h_i l_i = H$ and the fact that employment $l_i$ is the product of labour share $s_i$ times the population $l_i = s_i S_i$, we obtain that $p_i = \frac{\alpha \tilde{w}_i s_i S_i}{H}$. From optimal consumption $c_i, h_i$, the indirect utility can be written as $u_i = \alpha^\alpha (1-\alpha)^{1-\alpha} \frac{\tilde{w}_i}{p_i^\alpha}$. Using the expression for $p_i$ we can write $u_i = \varepsilon_i (1-\alpha)^{1-\alpha} \frac{\tilde{w}_i^{1-\alpha} H^\alpha}{(s_i S_i)^\alpha}$. Mobility between cities must equalize utility, so for any two cities $i$ and $j$, it must be the case that $u_i = u_j$ or $\varepsilon_i \frac{w_i^{(1-\alpha)(1-\tau)}}{(s_i S_i)^\alpha} = \varepsilon_j \frac{w_j^{(1-\alpha)(1-\tau)}}{(s_j S_j)^\alpha}$, where we have used the tax schedule to substitute for post tax wages $\tilde{w}$. Therefore in a country with $N$ cities, the location equilibrium is pinned down by $N$ first order conditions for production, $N-1$ conditions of free mobility, and 1 condition to allocate

the entire population to at least one city:

$$
\begin{aligned}
w_i &= \gamma A_i (s_i S_i)^{\gamma - 1} \\
\varepsilon_i \frac{w_i^{(1-\alpha)(1-\tau)}}{(s_i S_i)^{\alpha}} &= \varepsilon_1 \frac{w_1^{(1-\alpha)(1-\tau)}}{(s_1 S_1)^{\alpha}} \\
\sum_{i=1}^{N} S_i &= \mathcal{S},
\end{aligned}
\tag{2}
$$

where $\mathcal{S}$ is the overall population in the economy. We can solve this model explicitly when we substitute wages in the $N-1$ conditions for mobility where we normalize $\varepsilon_1$ to one. After rearranging we obtain for all $i = 2, \ldots, N$:

$$
S_i = S_1 \frac{s_1}{s_i} \left[ \varepsilon_i \left( \frac{A_i}{A_1} \right)^{(1-\alpha)(1-\tau)} \right]^{\frac{1}{(1-\gamma)(1-\alpha)(1-\tau)+\alpha}},
$$

or $S_i = K_i S_1$ where $K_i$ is a city-specific constant that depends on the technology and preference parameters $\gamma$ and $\alpha$, city-specific TFP $A_i$ and city-specific preferences $\varepsilon_i$, and the progressiveness of the tax system $\tau$. Since there is linear dependence of $S_i$ on $S_1$, we can explicitly solve this system using the last condition that $\sum_{i=1}^{N} S_i = \mathcal{S}$, or

$$
S_1 = \frac{\mathcal{S}}{\sum_{i=1}^{N} K_i},
$$

and $S_i = K_i S_1$ then gives us the solution for all $S_i$, $i = 2, \ldots, N$. We use the data on gross wages for each city $w_i$, the city population $S_i$, and the labor force share $s_i$. We set $\gamma = 1$, $\alpha = 0.3$, and $\tau = 0.2$. Then we can back out $A_i$ for each city from the first order condition for production, and the error terms $\varepsilon_i$ from the mobility condition. Now using the system of Eq (2) we can calculate the new equilibrium value $w_i^\star$, $S^\star$ for any tax schedule. We do this for $\tau = 0$, so that identical workers are treated equally in different cities (if workers were heterogeneous, taxes can be progressive, but city-specific). To obtain the energy consumption after the tax change is in effect, we use the relation between current energy consumption $E_i$ and the current wage bill $W_i = S_i w_i$: $E_i = \eta_i E_0 W_i^{0.83}$, where the coefficient 0.83 is obtained from the estimated elasticity. The term $\eta_i$ is obtained as the city-specific residual from the regression (1). Then given $\eta_i$ and the $W_i^\star$ we have obtained above, we can directly calculate $E_i^\star = \eta_i E_0 (W^\star)^{0.83}$. The change in the average energy consumption per unit of output produced then is $\dfrac{\sum_i \frac{E_i^\star}{W_i^\star}}{\sum_i \frac{E_i}{W_i}}$ (weighted by population). We use the same procedure for Waste to calculate $F_i^\star$.

**Results: Size-specific income tax.** Now we adjust the tax schedule by setting $\tau = 0$. Those in large cities will still pay more taxes because they earn higher wages, but they will not pay a higher average tax rate. In particular, the average tax rate will be the same for all workers in all cities. This makes the large cities relatively more attractive and the small cities relatively less attractive. As a result, there will be mobility of some workers from small cities to large cities to restore equilibrium. Those workers become more productive, but also house prices will rise in big cities. The implication is that workers moving to large cities will consume less housing. As a result of living in a bigger city, they also consume less energy and produce less waste.

We calculate the energy consumption and waste production after the introduction of the new taxation regime with $\tau = 0$, reported in Figs 9 and 10. Those living in large cities demand less energy because the cities have grown even larger, and people live in even smaller spaces.

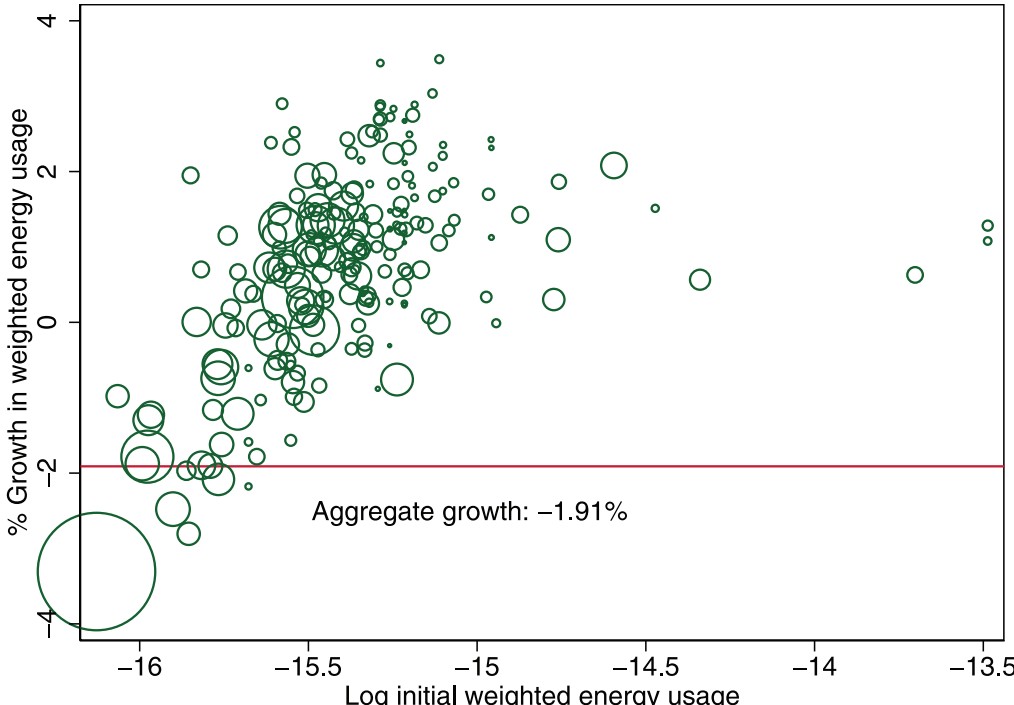

**Fig 9. Energy gain due to taxation change ($\tau = 0.2 \to \tau = 0$).** Taxation of labor income is changed from being progressive to flat with respect to a city's cost of living.

The smaller cities instead use more energy, because people have moved out. But more important is that the aggregate effect on energy demand is negative and more substantial than simply the average of all the cities' energy growth. The reason is that the population composition has changed. Many people have moved from small cities to large cities. It is precisely the movement of citizens from small cities to large cities that generates the energy gain. The aggregate energy gain is 1.91%. Likewise, the aggregate gain in terms of waste production is 1.87%. These environmental gains can be achieved without any penalty on economic output. While these gains are modest, we compare them to the change in total energy consumption in Great Britain in recent times. Total energy consumption in Great Britain between 2011 and 2016 fell by 3.4%, see the energy consumption statistics described in the data section, thus indicating that even a tax change, which does not impose a direct economic cost, could produce environmental gains of similar magnitude as Great Britain has seen in recent years. To further strengthen the applicability of this result, future work should determine to what extent other side effects of increased density, like air quality and urban heat islands, are taken into account in individuals location decisions. This is important, because deteriorating air quality affecting health outcomes and heat islands are of particular concern for urban environments, see [19, 27].

## Conclusion

Big cities look the opposite of the natural, unspoiled landscapes and little villages. Yet, living in those natural environments imposes a big environmental cost in terms of energy consumption and waste production. It is much more energy efficient to produce in densely populated cities rather than in scarcely populated rural towns. In this paper, we make three contributions. First, using novel data for Great Britain we show that the energy premium is 17% and the

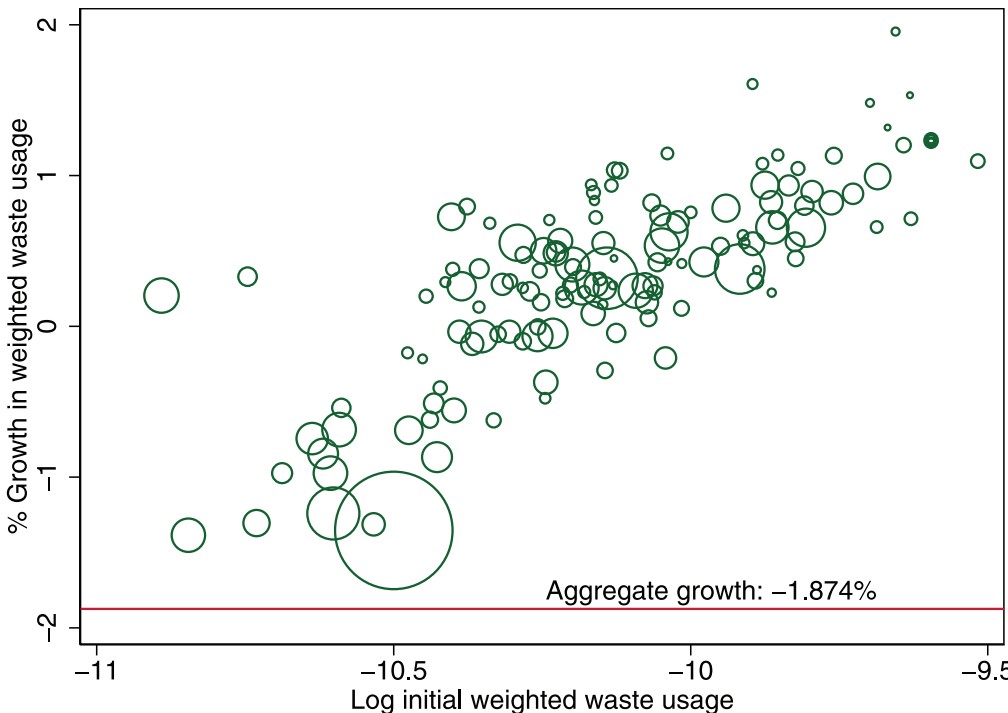

**Fig 10. Waste gain due to taxation change ($\tau = 0.2 \rightarrow \tau = 0$).** Taxation of labor income is changed from being progressive to flat with respect to a city's cost of living.

waste premium is 10%. Second, we build a model with population mobility that quantifies the relation between the output households produce, by choosing between urban and rural locations, and the associated energy consumption and waste generation. Third, we use the model to analyze an income taxation counterfactual that makes living in big, efficient cities relatively more desirable and therefore leads to energy efficiency gains by, in equilibrium, reallocating citizens towards more efficient cities. These gains can be achieved without lowering economic output. This is in contrast to most policies that induce a reduction of energy usage, because their effects are not unambiguously positive. Our results establish that policies towards denser living have the potential to be ecologically beneficial without introducing costly trade-offs.

## Acknowledgments

We would like to thank numerous colleagues and seminar audiences for insightful discussions and comments. Michael Amior collected the data and provided excellent research assistance.

## Author Contributions

**Conceptualization:** Jan Eeckhout, Christoph Hedtrich.

**Data curation:** Jan Eeckhout, Christoph Hedtrich.

**Formal analysis:** Jan Eeckhout, Christoph Hedtrich.

**Funding acquisition:** Jan Eeckhout.

**Investigation:** Jan Eeckhout, Christoph Hedtrich.

**Methodology:** Jan Eeckhout, Christoph Hedtrich.

**Project administration:** Jan Eeckhout.

**Resources:** Jan Eeckhout.

**Software:** Jan Eeckhout, Christoph Hedtrich.

**Supervision:** Jan Eeckhout.

**Validation:** Jan Eeckhout, Christoph Hedtrich.

**Visualization:** Jan Eeckhout, Christoph Hedtrich.

**Writing – original draft:** Jan Eeckhout.

**Writing – review & editing:** Jan Eeckhout, Christoph Hedtrich.

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
