## [Decision Letter · Decision Letter 0]

15 Mar 2021

PONE-D-20-27410

Green Urbanization

PLOS ONE

Dear Dr. Eeckhout,

Thank you for submitting your manuscript to PLOS ONE. After careful consideration, we feel that it has merit but does not fully meet PLOS ONE’s publication criteria as it currently stands and needs a careful **major revision**. Therefore, we invite you to submit a revised version of the manuscript that addresses the points raised during the review process.

We look forward to receiving your revised manuscript.

Kind regards,

Taoyuan Wei

Academic Editor

PLOS ONE

"X"

"X"

5. We note you have included a table to which you do not refer in the text of your manuscript. Please ensure that you refer to Table 2 in your text; if accepted, production will need this reference to link the reader to the Table.

Reviewers' comments:

Reviewer's Responses to Questions

**Comments to the Author**

1. Is the manuscript technically sound, and do the data support the conclusions?

Reviewer #1: Partly

Reviewer #2: Yes

Reviewer #3: Yes

2. Has the statistical analysis been performed appropriately and rigorously? 

Reviewer #1: No

Reviewer #2: Yes

Reviewer #3: Yes

3. Have the authors made all data underlying the findings in their manuscript fully available?

Reviewer #1: Yes

Reviewer #2: Yes

Reviewer #3: No

4. Is the manuscript presented in an intelligible fashion and written in standard English?

Reviewer #1: Yes

Reviewer #2: Yes

Reviewer #3: Yes

5. Review Comments to the Author

Reviewer #1: See attached file XXXXXXXXXXXXXXXXXXXXXXXXXXXXXXXXXXXXXXXXXXXXXXXXXXXXXXXXXXXXXXXXXXXXXXXXXXXXXXXXXXXXXXXXXXXXXXXXXXXXXXXXXXXXXxxXXXXXXXXXXXXXXXXXXXXXXXXXXXXXXXXXXXXXXXXXXXXXXXXXXXXXXXXXXXXXXXXXXXXXXXXXXXXXXXXXXXXXXXXXXXXXXXXXXXXXXXXXXXXXXXXXXXXXXXXXXXXXXXXXXX

Reviewer #2: Review comments

This manuscript provided new evidence on large cities and small towns in green urbanization. The topic of this paper is impressive; however, some expressions should be well-defined. Furthermore, the results' discussions are not extensive, and I have recommended the authors provide the sensitive analysis and comparisons with other studies. A minor revision is suggestive this time, and some comments may be helpful to the authors:

1. The motivation of your research is not clearly shown in the abstract. Some descriptions are not closely related not your topic.

2. In the Introduction, the authors need to further clearly state contributions and novelty in your paper.

3. The authors have reviewed many studies of green urbanization, but the research gap is not identified. Furthermore, the research motivation should be given with more details. It is suggested to review the related studies by contents, methods, and factors.

4. In the data section (Methods and Data), All data is for the 2009 cross-section; the author may need to update

5. The contribution to this energy premium of the industry is 12%(Results). Experience knows that industry consumes the most energy. Can we find the conclusions of previous studies, verify and explain this result in detail?

6. In terms of the article structure, I think it’s better to put the Methods and Data before the Results.

7. There are several repetitions between the conclusions and main contents, and it is recommended to summarize them with more refined sentences.

8. In the conclusions and summarizing the actions taken and results, please strengthen the explanation of their significance. It is recommended to use quantitative reasoning comparing with appropriate benchmarks, especially those stemming from previous work.

9. The paper should be proofread for better readability and correction of any grammar issue before publication.

Reviewer #3: The authors use various datasets to support the fact that large cities, not only more productive, but also more energy efficient and green. The paper is well written and well organized, and I think it could be accepted for publication after minor revision.

6. PLOS authors have the option to publish the peer review history of their article (what does this mean?). If published, this will include your full peer review and any attached files.

Reviewer #1: No

Reviewer #2: No

Reviewer #3: No

---

## [Author Response · Author response to Decision Letter 0]

6 Sep 2021

Responses to Reviewers are in a separate document that is uploaded with this revision

---

## [Decision Letter · Decision Letter 1]

21 Sep 2021

PONE-D-20-27410R1Green UrbanizationPLOS ONE

Dear Dr. Eeckhout,

Thank you for submitting your manuscript to PLOS ONE. After careful consideration, we feel that it has merit but does not fully meet PLOS ONE’s publication criteria as it currently stands. Therefore, we invite you to submit a revised version of the manuscript that addresses the points raised during the review process.

We look forward to receiving your revised manuscript.

Kind regards,

Taoyuan Wei

Academic Editor

PLOS ONE

Journal Requirements:

Additional Editor Comments (if provided):

The reviewers are almost satisfied with your revised version and a minor revision is required this time. Please carefully have a thorough check of your manuscript to make it perfect for publishing. Thank you!

Reviewers' comments:

Reviewer's Responses to Questions

**Comments to the Author**

1. If the authors have adequately addressed your comments raised in a previous round of review and you feel that this manuscript is now acceptable for publication, you may indicate that here to bypass the “Comments to the Author” section, enter your conflict of interest statement in the “Confidential to Editor” section, and submit your "Accept" recommendation.

Reviewer #1: (No Response)

Reviewer #2: All comments have been addressed

Reviewer #3: All comments have been addressed

2. Is the manuscript technically sound, and do the data support the conclusions?

Reviewer #1: Partly

Reviewer #2: Yes

Reviewer #3: Yes

3. Has the statistical analysis been performed appropriately and rigorously? 

Reviewer #1: Yes

Reviewer #2: Yes

Reviewer #3: Yes

4. Have the authors made all data underlying the findings in their manuscript fully available?

Reviewer #1: Yes

Reviewer #2: Yes

Reviewer #3: (No Response)

5. Is the manuscript presented in an intelligible fashion and written in standard English?

Reviewer #1: Yes

Reviewer #2: Yes

Reviewer #3: Yes

6. Review Comments to the Author

Reviewer #1: 2nd Round Review of “Green Urbanization” for PLoS ONE

The authors addressed my comments from the first round of reviews, and I have few remaining concerns, within the Journal’s framework for acceptability. They are as follows:

Major comment:

I am not yet satisfied with the author’s response regarding point 3d. they indicate that the balance of the factors which shape location decision agglomerate into the residual. While this is true, that doesn’t speak to the possibility of endogenous relationships that may exist with all those location-shaping factors, or how much insight we are actually gaining through their variable of interest.

While I won’t debate the quality of the model’s explanatory power, I do take issue with lack of transparency into this for the reader. Throughout this paper, the authors provide only evidence of loadings on their variables of interest, with no insight into how well the model fits the question, or the explanatory power (level and consistency of results) of any included control variables. There should be regression tables included so the reader can see the details of your model’s fit and overall explanatory power.

Minor comments:

1) Introduction, paragraph 2. The last sentence is unclear, and could be misinterpreted by the reader. Please clarify, and cite.

2) The first few sentences under “The Urban Wage Premium” require citation(s).

3) Same section, the authors use the terms “total population” and “population size.” I believe these are meant to be interchangeable references to the same data measure, but the authors should choose one term and use it consistently to avoid confusion by the reader.

Reviewer #2: The methodological approach responds to the minimum requirements usually established and the results obtained are of special interest to science. The bibliography is extensive and sufficiently related. A revision in grammatical forms is suggested. I resolve to positively rate the article presented.

Reviewer #3: (No Response)

7. PLOS authors have the option to publish the peer review history of their article (what does this mean?). If published, this will include your full peer review and any attached files.

Reviewer #1: No

Reviewer #2: No

Reviewer #3: **Yes: **Chengzheng Li

---

## [Author Response · Author response to Decision Letter 1]

4 Nov 2021

A separate document with detailed responses is uploaded

---

## [Editor Report · Decision Letter 2]

10 Nov 2021

Green Urbanization

PONE-D-20-27410R2

Dear Dr. Eeckhout,

We’re pleased to inform you that your manuscript has been judged scientifically suitable for publication and will be formally accepted for publication once it meets all outstanding technical requirements.

Kind regards,

Taoyuan Wei

Academic Editor

PLOS ONE

---

## [Editor Report · Acceptance letter]

17 Nov 2021

PONE-D-20-27410R2 

Green Urbanization 

Dear Dr. Eeckhout:

I'm pleased to inform you that your manuscript has been deemed suitable for publication in PLOS ONE. Congratulations! Your manuscript is now with our production department. 

Kind regards, 

on behalf of

Dr. Taoyuan Wei 

Academic Editor

PLOS ONE